Alpha-actinin of the chlorarchiniophyte Bigelowiella natans

Backman Lars lars.backman@chem.umu.se
Department of Chemistry, Umeå University , Umeå , Sweden
Ettrich Rüdiger
Electronic publication date: 2018 Jan 17
Publication date: 2018
Volume: 6
Electronic Location ID: e4288
Received 2017 Oct 9; Accepted 2018 Jan 3
Copyright: ©2018 Backman
Copyright year: 2018
Copyright holder: Backman
License: This is an open access article distributed under the terms of the Creative Commons Attribution License, which permits unrestricted use, distribution, reproduction and adaptation in any medium and for any purpose provided that it is properly attributed. For attribution, the original author(s), title, publication source (PeerJ) and either DOI or URL of the article must be cited.
License URL: https://creativecommons.org/licenses/by/4.0/

Keywords: Spectrin repeat, Actin-binding protein, Bigelowiella natans, Alpha-actinin, Calcium-binding protein

Funding: Carl Tryggers Stiftelse CTS 13:31 This work was supported by grants from Carl Tryggers Stiftelse (CTS 13:31). The funders had no role in study design, data collection and analysis, decision to publish, or preparation of the manuscript.

==============================
The genome of the chlorarchiniophyte Bigelowiella natans codes for a protein annotated as an α-actinin-like protein. Analysis of the primary sequence indicate that this protein has the same domain structure as other α-actinins, a N-terminal actin-binding domain and a C-terminal calmodulin-like domain. These two domains are connected by a short rod domain, albeit long enough to form a single spectrin repeat. To analyse the functional properties of this protein, the full-length protein as well as the separate domains were cloned and isolated. Characerisation showed that the protein is capable of cross-linking actin filaments into dense bundles, probably due to dimer formation. Similar to human α-actinin, calcium-binding occurs to the most N-terminal EF-hand motif in the calmodulin-like C-terminal domain. The results indicate that this Bigelowiella protein is a proper α-actinin, with all common characteristics of a typical α-actinin.

Introduction

The cellular cytoskeleton is indispensable for any eukaryotic cell. This protein network, composed to varying degrees of actin filaments, intermediate filaments and microtubules, plays a role in all cellular events. The organisation and dynamics of the actin cytoskeleton is regulated by a variety of actin-binding proteins and other accessory proteins (Dos Remedios et al., 2003; McGough, 1998; Winder & Ayscough, 2005). One such accessory protein is α-actinin, that due to its ability to form antiparallel dimers is capable to cross-link actin filament into fibers or bundles as well as networks (Foley & Young, 2014; Wachsstock, Schwarz & Pollard, 1993).

α-actinin, like other members of the spectrin superfamily, is characterised by three structural domains: a N-terminal actin-binding domain, composed of two calponin homology domains, and a C-terminal calmodulin-like domain (Broderick & Winder, 2005; Wasenius et al., 1987). These domains are connected by a central rod domain consisting of spectrin-like repeats (Blanchard, Ohanian & Critchley, 1989; Otey & Carpen, 2004; Sjöblom, Salmazo & Djinovic-Carugo, 2008). Depending on the source of α-actinin, the rod domain contains one, two or four spectrin repeats. Vertebrate and invertebrate α-actinins have a rod with four repeats whereas α-actinin of fungal origin contains only two repeats (Virel & Backman, 2004; Virel & Backman, 2007). Some protozoa, like Entamoeba histolytica and Encephalitozoon cuniculi have a much shorter rod sequence, that may form a single repeat but more likely fold into a coiled-coil region (Virel & Backman, 2004). Plants, algea and other photosynthesising organisms seem to lack α-actinin or α-actinin-like proteins. Therefore, it was surprising that the genome of the photosyntesising chlorarachniophyte Bigelowiella natans (Neilson, Rangsrikitphoti & Durnford, 2017) contains a gene that is annotated as a α-actinin-like protein (Curtis et al., 2012). No other members of the spectrin superfamily appears to be present in Bigelowiella.

Chlorarachniophytes are believed to have arisen through a secondary endosymbiosis when a green algae (a chlorophyte) was engulfed by a non-photosyntesising eukaryotic host (Archibald et al., 2003; Curtis et al., 2012; Gould, Waller & McFadden, 2008; Neilson, Rangsrikitphoti & Durnford, 2017; Rogers et al., 2004). Chlorarachniophytes have genetic material at four distinct locations; in the nucleus, mitochondria, plastid and nucleomorph, the remnant genome of the engulfed green algae (Curtis et al., 2012). Only a small number of genes (331) remains in the nucelomorph as most have been transferred to the nucleus or lost (Gilson et al., 2006). As present day green algaes lack α-actinin or α-actinin-like proteins, it is likely that the gene was already present in the genome of the host.

According to the JGI genome portal, the genome of Bigelowiella contain 610 gene models considered to be cytoskeletal. In addition to α-actinin, homologues of common actin-binding proteins are present; such as Arp2/3, filamin, gelsolin and many more. Recently, the evolution of key elements of the Chlorarachniophyte cytoskeleton in relation to the other Rhizaria phyla Foraminifera and Radiolariain were described (Krabberød et al., 2017). It was suggested that pseudopodia of Chlorarachniophytes rely on actin structures whereas those of Foraminifera and Radiolariain are dependent on microtubules. In vertebrates, α-actinin is involved in the attachment of actin filaments to membranes as well as in filopodia and lamellipodia formation (Geiger et al., 1984; Hamill et al., 2015; Sobue & Kanda, 1989; Wehland, Osborn & Weber, 1979). If the Bigelowiella α-actinin-like protein is a genuine α-actinin, it can be expected to have similar functions also in this organism. To investigate this, the Bigelowiella α-actinin-like protein and its structural domains were cloned, expressed in bacteria, isolated and characterized.

To my knowledge this is the first report on cloning, expression, isolation and characterisation of a Bigelowiella protein.

Materials and Methods

Cloning, expression and purification

The sequence of B. natans α-actinin was obtained from Joint Genome Institute (JGI), accession number 89463. To facilitate subcloning, the N-terminal methionine was removed and BamHI and XhoI restriction sites were added to the 5′- and 3′-end, respectively. The final gene was synthesized and inserted in plasmid pUC57 by Genscript (Piscataway, NJ, USA). The resulting plasmid pUC57-BigN-ACTN was used to transform (by heat shock) competent E. coli TG1 cells. After over-night culture at 37 °C in Luria-Bertani medium containing 100 µM carbenicillin, plasmids were isolated using QIAprep Spin miniprep kit (Qiagen GmbH, Hilden, Germany). The α-actinin gene was excised by Bam HI and XhoI, gel purified and ligated into pET-TEV (a modified pET-19b vector) containing an N-terminal 10xHis-tag and a TEV protease cleavage site. The resulting plasmid pTEV-BigN-ACTN was then used to transform (by heat shock) competent E. coli BL21(DE3).

Breakpoints for the structural domains of B. natans α-actinin-like protein were determined by alignment with other α-actinins as well as the database Superfamily (Wilson et al., 2009). When Smart (Letunic, Doerks & Bork, 2015) and Pfam (Finn et al., 2016) were interrogated with the amino acid sequence of B. natans α-actinin-like protein similar domain assignments were returned (Table 1). As shown in Fig. 1, the expressed full length B. natans α-actinin-like protein contains residues 2 to 524. The actin-binding domain (ABD) spans residues 2 to 260, the actin-binding domain with the rod domain (ABD-ROD) spans residues 2 to 372 and the rod domain (ROD) spans residues 257–392. Two constructs of the calmodulin-like domain (EF) were made, the longer construct (EF) contains residues 326 to 524 and the shorter (short-EF) begin at residue 370.

Table 1 Domain assignments of Bigelowiella natansα-actinin by Superfamily, Smart and Pfam.

	Superfamily	Smart	Pfam	
	Region	E-value	Region	E-value	Region	E-value	
Calponin homology domain	14–254	9.75e–56	26–131	5.82e–12	24–134	6.6e–16	
Calponin homology domain	–	–	148–250	1.84e–23	146–256	9.0e–25	
Spectrin repeat	274–384	3.7e—17	279–392	5.2e–2	–	–	
EF-hand	372–488	9.45e–14	392–421	4.4e–6	392–420	2.6e–6	
Ca2 +-insensitive EF-hand			459–520	1.44e–8	459–520	4.1e–9	
Notes.

The amino acid sequence of Bigelowiella natans a-actinin was submitted to Superfamily (Wilson et al., 2009), Smart (Letunic, Doerks & Bork, 2015) and Pfam (Finn et al., 2016).

Figure 1 Domain organization of Bigelowiellaα-actinin-like protein.

The full-length protein contains a N-terminal actin-binding domain (ABD), a central rod domain (ROD), composed of one spectrin repeat and a C-terminal calmodulin-like domain (EF and shortEF) with EF-hand motifs. The first amino acid residue (methionine) was removed in all constructs.

Plasmids expressing ABD, ABD-ROD, ROD and short-EF were created by mutagenesis of pTEV-BigN-ACTN. The gene fragment coding for EF was synthesized by Genscript, and ligated into pET-TEV, as described above, creating plasmid pTEV-BigN-EF.

Gene fragments were also inserted in pGEX-6P-2, adding glutathione S-transferse (GST) to the N-terminal instead of the His-tag. This resulted in plasmids pGEX-BigN-ABD, pGEX-BigN-ABD-ROD and pGEX-BigN-EF. To improve purification of expressed proteins, a His-tag (with 6 His residues) were added to the C-terminal. All constructs were sequenced (Eurofins MWG GmbH, Ebersberg, Germany; Genscript, Piscataway, NJ, USA) to control the fidelity of the final clones.

E. coli BL21(DE3) cells were transformed (by heat-shock) with the purified plasmids containing the different constructs. The transformed cells were cultured at 37 °C in Luria-Bertani medium containing 100 µM carbenicillin until an optical density of 0.6–0.8 at 600 nm was reached. Protein expression was induced by addition of isopropyl thio- β-D-galactoside to a final concentration of 0.5 mM and cells were grown overnight at 16 °C. Cells were harvested by centrifugation (29,000× g for 15 min), resuspended in 25 mM sodium phosphate buffer, pH 7.6, 150 mM NaCl and stored at −20 °C until the expressed protein was purified.

For purification, cells were thawed, polyethyleneimine was added to a final concentration of 0.05% and then lysed by sonication on ice. One-tenth volume of 10% Triton X-100 was added to the lysed cells. After 30 min incubation on ice, cell debris was removed by centrifugation at 37,000× g for 20 min and the clarified supernatant loaded on an appropriate column.

Clarified supernatant of N-terminal His-tagged proteins were loaded on HiTrap™ Chelating HP (GE Healthcare Bioscience AB, Sweden) columns charged with nickel. Unbound proteins were eluted with 25 mM sodium phosphate buffer, pH 7.6, 150 mM NaCl, 150 mM imidazole. Bound proteins were eluted with an imidazole gradient ranging from 150 to 500 mM imidazole in the same buffer. Imidazole was removed by gel filtration on HiPrep desalting columns (GE Healthcare Bioscience AB, Stockholm, Sweden). When required, Tobacco Etch Virus (TEV) protease (kindly provided by Dr. David S. Waugh) was used to hydrolyse the His-tag. The released 10xHis-tag and the 6xHis-tagged TEV protease were removed by affinity chromatography on a nickel-charged HiTrap™ Chelating HP as before. To improve solubility, purified proteins were transferred into buffer TK (50 mM Tris, 200 mM KCl, pH 8.0) or buffer TKMG (50 mM Tris, 200 mM KCl, 10 mM β-mercaptoethanol, 10% glycerol, pH 8.0) by gel filtration on HiPrep desalting columns.

GST-tagged proteins were loaded on Glutathione-Sepharose columns (GE Healthcare Bioscience AB, Stockholm, Sweden). After unbound proteins had been eluted, bound proteins were eluted with 20–30 mM glutathione in 50 mM Tris, pH 8.0. The GST-tag was liberated by overnight incubation in the presence of protease C, obtained from the Protein Expert Platform (Umeå, Sweden). The GST-tagged moiety was removed by a second passage over the Glutathione-Sepharose. As before, purified proteins were transferred into buffer TK or TKMG by gel filtration on HiPrep desalting columns.

GST-free proteins with a C-terminal His-tag were purified further using affinity chromatography on a nickel-charged HiTrap™ Chelating HP column as described above. The isolated protein was finally transferred into buffer TK or TKMG by gel filtration on a Hiprep 26/10 desalting column.

Protein concentration was determined from the absorbance at 280 nm using the molar absorptivity, as calculated from the amino acid sequence (using ProtParam at the ExPASy proteomics server). The purity of the expressed polypeptides was routinely determined under denaturating conditions by SDS-polyacrylamide gel electrophoresis (Laemmli, 1970).

Gel filtration

The propensity of purified polypeptides to form dimers or aggregates were determined by gel filtration on Toyopearl HW50-S 10/300 and Superdex 200 columns. Elution profiles were determined both in buffers TK and TKMG. Since the presence of glycerol affected the elution behaviour, a small volume of dissolved tryptophan was added to each sample as an internal standard and elution profiles were adjusted accordingly.

Ferritin (440 kDa) and bovine serum albumin (67 and 135 kDa) was used as references.

Actin co-sedimentation assay

Co-sedimentation was used to assay the ability of Bigelowiella α-actinin-like protein to cross-link actin. Actin was purified from rabbit skeletal muscle acetone powder as before (Pardee & Spudich, 1982; Spudich & Watt, 1971). Monomeric actin, in 5 mM Tris-HCl, pH 8.0, 0.2 mM CaCl2 and 0.2 mM ATP, was polymerised by addition of KCl and MgCl2 to final concentrations of 50 mM and 2 mM, respectively. Actin was allowed to polymerise for 60 min at room temperature. Polymerised actin was mixed with varying concentrations of full-length Bigelowiella α-actinin-like protein and incubated for 30 min at room temperature. After incubation, the reaction mixture was centrifuged at 13,000 rpm (16,000× g) for 15 min and supernatants and pellets were separated and analysed by SDS-PAGE (Laemmli, 1970). The amounts of actin in the pellets were quantified using Image Lab 5.2 (Bio-Rad laboratories).

Negative staining electron microscopy

The samples assayed by the co-sedimentation were also analysed by electron microscopy. Copper grids coated with formvar and carbon were prepared with Leica EM ACE200 carbon coating system. Grids were glow-discharged with Pelco easiGlow system (Ted Pella, Inc., Redding, CA, USA). 3.5 µl sample adsorbed for 2 min, washed twice in water and immediately stained in 50 µl 1.5% uranyl acetate solution for 30 s. Samples were examined with Talos 120C (FEI, Eindhoven, The Netherlands) operating at 120 kV. Micrographs were acquired with a Ceta 16M CCD camera (FEI, Eindhoven, The Netherlands) using TEM Image & Analysis software ver. 4.15 (FEI, Eindhoven, The Netherlands).

Circular dichroism (CD) spectroscopy

Expressed polypeptides in buffer TK or TKMG with or without urea were analysed by CD spectroscopy using a Jasco J-810 spectrometer. Spectra between 200 and 260 nm were collected using 0.025 nm step-size and a scan speed of 50 nm per min, with a response time of 0.5 s and a bandwidth of 1 nm. Mean residue molar ellipticity was calculated from three accumulated spectra.

The temperature stability of expressed polypeptides was determined by the ellipticity at 222 nm. The thermal scan rate was 1 °C/min, with a data pitch of 0.2 °C, 2 nm bandwidth and 4 s response time.

Structure prediction

RaptorX (Kallberg et al., 2012; Ma et al., 2013) were used to predict the tertiary structure of separate domains. UCSF Chimera package was used to analyse structures and draw figures (Pettersen et al., 2004).

Calcium assay

45Ca autoradiography was used to determine calcium binding as described (Maruyama, Mikawa & Ebashi, 1984). Bigelowiella α-actinin-like protein was blotted onto a PDVF membrane, wetted with 60 mM KCl, 5 mM MgCl2, and 10 mM imidazole-HCl pH 6.8. The membrane was then incubated in the same solution containing ∼50 µM 45CaCl2 (20.87 mCi/mg) for 1 h, rinsed with buffer and dried. Autoradiography was used to detect radioactive calcium.

Results and Discussion

Cloning, expression and purification

The amino acid sequence of Bigelowiella α-actinin-like protein was retrieved from JGI. Based on this amino acid sequence corresponding DNA was synthesised. After subcloning into the expression vector (pET-TEV), this plasmid was used as template for preparing constructs of polypeptides used in this report.

All constructs expressed well and could be purified by metal chelating chromatography to reasonable purity. However, after removing the imidazole used to release the polypeptides from the metal chelating column, only the two EF peptides were soluble in salt-free buffer (50 mM Tris, pH 8.0). All other polypeptides as well as the full-length protein required addition of salt (NaCl and/or KCl) to reduce precipitation and aggregation. Although inclusion of 10% glycerol increased solubility of ABD, still a fraction, that increased with time, precipitated. Independent on the buffer systems tested, it has not been possible to find conditions to keep the polypeptides containing the rod domain, ABD-ROD and ROD, in solution.

Full-length a-actinin-like protein

After the full-length protein was released from the nickel-column and transferred into a imidazole-free buffer, the protein begun to form a visible precipitate. Inclusion of high concentration KCl together with glycerol and β-mercaptoethanol improved solubility but still a significant fraction was in an aggregated form.

By including a high-speed centrifugation step (343,000× g for 30 min) to remove aggregates before buffer change, no precipitation or aggregation was observed after change to the TKMG buffer. Although soluble, the amount of full-length protein that remained in solution decreased considerable and it degraded with time. When this material was applied on a Superdex 200 gel filtration column, the full-length protein eluted at a position between monomeric and dimeric bovine serum albumin, indicating that no higher oligomers or aggregates were present (Fig. 2A). The peak that eluted at ca 18 ml (seen as a shoulder in the figure), contained degradation products as seen in Fig. 2B.

Figure 2 Gel filtration of Bigelowiellaα-actinin-like protein.

(A) The oligomeric state of the full-length α-actinin-like protein was determined by gel filtration on a Superdex 200 column, equilibrated with 50 mM Tris, 200 mM KCl, 10 mM β-mercaptoethanol, 10% glycerol, pH 8.0. 0.1 ml His-tagged protein was applied after centrifugation at 343,000× g for 30 min. The flow rate was 0.5 ml per min and absorbance at 280 was measure continuously. Ferritin (440 kDa), monomeric (67 kDa) and dimeric (135 kDa) bovine serum albumin were used as references. (B) Eluted fractions (0.5 ml) were collected and analysed by gel electrophoresis. From left to right, the stained gel shows protein content of fractions eluted between 14 and 20 ml.

ABD

Similar to the full-length protein, also ABD displayed a propensity to precipitate when imidazole was removed. Although inclusion of 10% glycerol and β-mercaptoethanol increased solubility of ABD, still a fraction, that increased with time, precipitated. When aggregated material was removed by high-speed centrifugation (343,000× g for 30 min), monomeric ABD remained in the supernatant as determined by gel filtration. However, upon storage new aggregates formed as shown in Fig. 3.

Figure 3 Gel filtration of Bigelowiella actin-binding domain (ABD).

Aggregation was analysed by gel filtration on a Toyopearl HW50-S 10/300 column, equilibrated with 50 mM Tris, 200 mM KCl, 10 mM β-mercaptoethanol, 10% glycerol, pH 8.0. 0.1 ml His-tagged protein was applied directly after isolation (red) or after centrifugation at 343,000× g for 30 min (blue). The flow rate was 0.5 ml per min and absorbance at 280 nm was measured continuously. Void volume was determined from the elution pattern of ferritin (440 kDa). Data shown is representative of three independent experiments.

ROD and ABD-ROD

It was not possible to find conditions to keep either of these polypeptides in solution after the initial purification step. Therefore, the properties of these two could not be determined reliably.

EF and short-EF

EF and short-EF could be purified to homogeneity even after cleavage of the N-terminal His-tag used for the initial purification step. Both EF-and short-EF were monomeric even in the absence of salt as determined by gel filtration.

Calcium binding

Initially a 45Ca-overlay assay showed that the Bigelowiella full-length α-actinin-like protein has affinity for calcium ions as shown in Fig. 4. The band intensity indicated that the binding affinity is similar to that of Entamoeba histolytica α-actinin2 (Virel, Addario & Backman, 2007). To analyse the calcium binding further a CD assay was used. The rationale being that ligand binding will increase the melting temperature of any protein, as the ligand must be dissociated before the protein unfolds and this require more energy, i.e., heat in this case.

Figure 4 Calcium-binding properties of Bigelowiellaα-actinin-like protein.

Bigelowiella α-actinin-like protein was slot-blotted onto a PVDF membrane, together with Entamoeba α-actinin1 (positive control) and S. pombe α-actinin (negative control). Calcium-binding was probed by a calcium-overlay assay using 45Ca.

Figure 5 Far UV-CD spectroscopy.

(A) Far UV-CD spectra of the calmodulin-like C-terminal of Bigelowiella in the absence (red) and presence (blue) 10 mM calcium ions. The mean residue molar ellipticity was determined from three accumulated scans between 200 and 260 nm at 20 °C, in 50 mM Tris, pH 8.0. (B) Urea unfolding of the calmodulin-like C-terminal in the absence (red) and presence (blue) of 10 mM calcium ions. The mean residue molar ellipiticy at 222 nm was determined from three accumulated scans between 200 and 260 nm at 20 °C, in 50 mM Tris, pH 8.0. (C) The terminal stability of the calmodulin-like C-terminal in the absence (red) and presence (blue) of 10 mM calcium ions in 50 mM Tris, pH 8.0 and 2.9 M urea was determined from the mean residue molar ellipticity at 222 nm.

In all buffers tested, the CD spectra of EF were typical of a folded helical protein, with the typical negative peaks at 208 and 222 nm, independent on whether calcium ions were present or not (Fig. 5A). When comparing the melting traces in the absence and presence of calcium of EF and short-EF, only negligible differences were noticeable. It also was evident that the C-terminal domain was highly stable even at high temperatures. Therefore, in order to destabilize the protein, the denaturant urea was included in the CD measurements. As shown in Fig. 5B, increasing concentrations of urea led to increased unfolding of EF in the absence of calcium ions. However, in the presence of calcium ions, EF unfolded only partly. When the melting experiments were repeated in the presence of urea (∼3 M), it was evident that calcium stabilised EF; in the absence of calcium the melting transition was around 50 °C whereas addition of calcium increased the transition to around 80 °C or even higher (Fig. 5C). The same results were obtained for shortEF (not shown). These results indicate that the calcium affinity is associated with the calmodulin-like C-terminal and the EF-hand motifs.

Superfamily (Wilson et al., 2009) identified residues 372–488 as an EF-hand domain with a very low E-value (4.84 × 10−15). Pfam (Finn et al., 2016) and Smart (Letunic, Doerks & Bork, 2015) identified residues 392–420 as an active EF-hand domain (E-values: 2.6 × 10−6 and 6.54 × 10−6, respectively) and residues 459–520 as a calcium-insensitive EF-hand domain (E-values: 4.1 × 10−9 and 1.44 × 10−8). Inspection and alignment to other EF-hand domains and the consensus sequence indicated that only the first EF-hand motif has proper residues in the positions involved in coordinating the calcium ion as Fig. 6 shows. The other three motifs lack one or more of the residues believed to be required for calcium binding.

Figure 6 Calcium binding motifs.

The sequence of the calmodulin-like C-terminal domain of Bigelowiella a-actinin-like protein was aligned with C-terminal domains of other α-actinins as well as with the canonical sequence. In the calcium binding loop residues at positions X, Y, Z, y, x and z coordinate the calcium ion, generally by side chain oxygens. The residue at position y coordinates the calcium ion through the backbone carbonyl oxygen. In the canonical sequence: E, glutamate; G, glycine; I, isoleucine, leucine or valine; n, hydrophobic residue; *, any residue. Based on the Prosite PS00018 pattern: D-W-[DNS]-ILVFYW-[DENSTG]-[DNQGHRK]-GP-[LIVMC]-[DENQSTAGC]-x(2)-[DE]-[LIVNFYM], red indicates a residue present in active EF-hand motifs, whereas blue indicates a residue not commonly found in this position of the calcium binding loop.

Actin binding

The ability of α-actinins to form antiparallel dimers creates binding sites for actin at each end of the dimer, that allows the dimer to cross-link or bundle actin filaments. It is generally believed that the rod domain is required for dimer formation and that the calponin homology domains constitute the binding site for actin. If correct also for the Bigelowiella α-actinin-like protein, ABD and ABD-ROD would bind actin filaments but only ABD-ROD would cross-link or bundle filaments, in addition to the full-length protein.

The rationale of the co-sedimentaion assay is that cross-linked or bundled actin filaments are pelleted by a low-speed centrifugation whereas any actin binder would only be pelleted with actin filaments by a high-speed centrifugation. Therefore, increasing concentrations of the Bigelowiella α-actinin-like protein were incubated with actin filaments before a low-speed centrifugation. As seen in Fig. 7A, in the absence of the α-actinin-like protein only a very small amount of actin was pelleted. On the other hand, in the presence of increasing concentrations of the full-length α-actinin-like protein, increasing amounts of both proteins were found in the pelleted material, indicating cross-linking or bundling of actin. Quantification indicated that a ratio of one cross-linker to two actin monomers were enough for maximal cross-linking (Fig. 7B), very similar to the behaviour of Entamoeba histolytica α-actinin (Virel, Addario & Backman, 2007). As gel filtration indicated that no higher oligomers were present, these results clearly indicate that the α-actinin-like protein dimerises.

Figure 7 Low speed actin co-sedimentation assay.

(A) Bigelowiella α-actinin-like protein and polymerised actin were incubated for 30 min separately or together at different concentrations. After 30 min at room temperature, the samples were centrifuged (13,000 rpm, 15 min) and supernatant (S) and pellet (P) were separated on 10% SDS-PAGE and visualized by Coomassie Blue staining. Lane 1: 5 µM actin; lanes 2–7: 5 µM actin together with 2.5, 1, 0.5, 0.2, 0.1 and 0.05 µM α-actinin-like protein; M: protein ladder. Data shown are representative of three experiments. (B) The fraction of actin pelleted by the presence of full-length α-actinin-like protein was quantified by analysis of stained gels using Image Lab.

This was corroborated by transmission electron microscopy. In the presence of the Bigelowiella protein actin filaments form networks and bundles, not seen in its absence (Fig. 8).

Figure 8 Electron transmission microscopy.

5.0 µM actin was incubated alone (A) or with 1.25 µM Bigelowiella a-actinin-like protein (B) as before, added to grids and negatively stained with uranylacetate. Bar: 200 nm.

Due to the propensity of ABD and ABD-ROD to aggregate and precipitate it was not feasible to investigate their binding further, as even a low-speed spin pelleted most of the ABD-ROD. Likewise, nearly all ABD was found in the pellet after a high-speed centrifugation.

Table 2 Validation scores for the predicted structures using Protein Structure Validation Software.

	ABD	ROD	EF	short EF	
	Mean score	Z-score	Mean score	Z-score	Mean score	Z-score	Mean score	Z-score	
Structure quality factors									
Procheck G-factor (phi/psi only)	0.21	1.14	0.57	2.56	0.07	0.59	0.02	0.39	
Procheck G-factor (all dihedral angles)	0.05	0.30	0.35	2.07	−0.04	−0.24	−0.01	−0.06	
Verify3D	0.45	−0.16	0.31	−2.41	0.24	−3.53	0.30	−2.57	
ProsaII	0.55	−0.41	0.78	0.54			0.82	0.70	
MolProbity clashscore	124.94	−19.91	83.22	−12.76	119.69	−19.01	96.20	−14.98	
Ramachandran plot summary from procheck									
Most favoured regions	89.9%		91.1%		86.5%		88.9%		
Additionally allowed regions	7.9%		8.1%		10.7%		9.7%		
Generously allowed regions	0.9%		0.8%		2.2%		0.0%		
Disallowed regions	1.3%		0.0%		0.6%		1.4%		
Ramachandran plot statistics from Richardson’s lab									
Most favoured regions	95.2%		94.9%		93.9%		92.3%		
Allowed regions	2.4%		4.4%		4.6%		5.8%		
Disallowed regions	2.4%		0.7%		1.5%		1.9%		

Homology modelling

The amino acid sequences of ABD, ROD, EF and short EF were submitted to the web server RaptorX for structure predictions (Kallberg et al., 2012; Ma et al., 2013). The qualities of the predicted structures were validated using the Protein Structure Validation Software suite (Bhattacharya, Tejero & Montelione, 2007). Table 2 shows validation scores for the predicted structures. The scores indicate that the structures are reasonable accurate, with only a few residues in disallowed regions. This is illustrated in Fig. 9, by the Ramachandran plot of the ABD model which places six residues in disallowed regions (Lys38, Val40, Pro42, Thr14, Pro165 and Val168).

Figure 9 Ramachandran plot of the actin-binding domain (ABD) of Bigelowiellaα-actinin-like protein.

The quality of the predicted structure of ABD was analysed by Rampage. Six residues (Lys38, Tyr40, Pro42, Thr144, Pro 165 and Val168) were found in disallowed regions.

One of the templates used in the homology modelling of ABD, was the actin-binding domain of Schizosaccharomyces pombe α-actinin. Although the sequence of Bigelowiella ABD is only ca 39% identical to the ABD of S pombe (pdb: 5bvr), the DaliLite server (Hasegawa & Holm, 2009) returns a very high Z-score of 32.5 and a low root mean deviation value of 1.0 Å. When the Dali server was interrogated with Bigelowiella ABD as query structure, several structures with very high Z-scores and low rmsd values were returned. The returned Z-score of the actin-binding domain of human α-actinin3 (pdb: 1wku) and α-actinin4 (pdb: 2r0o) were 33.8 and 33.5, respectively, and the rmsd were in both cases 1.0 Å. The major structural differences between the predicted structure of Bigelowiella ABD and determined structure of S pombe ABD are located to loops connecting the helices (Fig. 10).

The predicted structure of the Bigelowiella ROD domain is similar to all other determined spectrin repeats although the sequence identity in general is low (Fig. 11). When the amino acid sequence is compared to one of the templates used in the prediction, the rod domain of human α-actinin (pdb: 1hci), the sequence identity of either of the four repeats is around 20% or less (Fig. 12). Still the Z-scores and rmsd values range from 11.7 to 14.3 and 1.6 to 2.5, respectively, as determined by the DaliLite server.

Figure 10 The actin-binding domain.

The predicted structure of actin-binding domain (ABD) of Bigelowiella α-actinin-like protein (goldenrod) was superimposed with the crystal structure of S. pombe α-actinin ABD (blue, pdb: 5bvr).

Figure 11 The rod domain.

The model of the rod domain of Bigelowiella α-actinin-like protein (goldenrod) was superimposed with the first spectrin repeat of human α-actinin2 (blue, pdb: 1hci). The characteristic tryptophan important for the stability of spectrin repeats is shown explicitly.

Figure 12 Alignment of the rod domain.

The sequences of human (accession: NM_001130004), Entamoeba histolytica (accessions: AF208390 and XM_648191) and Schizosaccharomyces pombe (accession: NM_001019718) α-actinins were aligned with Bigelowiella natans α-actinin using Multalin (Corpet, 1988). Shaded regions are predicted by J Pred to be helical (Drozdetskiy et al., 2015).

Spectrin repeats are found in several cytoskeletal proteins, such as spectrin, dystrophin and plectin as well as α-actinins. The common structure of a spectrin repeat is a three-helix bundle. These repeats are defined by a tryptophan residue at position 17 in the first helix and a leucine residue two residues from the C-terminal end of the third helix. This tryptophan is believed to be essential for folding and stability of the triple-helix motif (Kusunoki, MacDonald & Mondragon, 2004; MacDonald et al., 1994; Pantazatos & MacDonald, 1997). Interestingly, such a tryptophan is present in the predicted ROD structure (Fig. 11).

Figure 13 The calmodulin-like C-terminal domains.

The predicted structure of the calmodulin-like C-terminal domain (goldenrod) of Bigelowiella α-actinin-like protein was superimposed with the N-terminal (blue) and C-terminal (red) halves, respectively, of E. histolytica (pdb: 2m7l).

The central domain of Entamoeba histolytica α-actinin1 is also short, spanning ca 120 residues, and probably does not form a spectrin repeat but rather a coiled structure (Virel & Backman, 2006). Since Entamoeba α-actinin1, like Bigelowiella α-actinin, cross-links actin filaments these α-actinins presumably dimerises by homotypic interactions between the central domain whether it is a spectrin repeat or not. In contrast, in α-actinins with four spectrin repeats dimerization occurs by heterotypic interactions between the first and last and between the second and third spectrin repeat (Djinovic-Carugo et al., 1999; Flood et al., 1995; Flood et al., 1997).

When the predicted structure of shortEF was submitted to the Dali server, the returned top-scorer was the C-terminal calmodulin-like domain of Entamoeba histolytica α-actinin2 (pdb: 2m7l), with a Z-score of 18.0 and a rmsd of 0.9 Å (Fig. 13). The same structure was also found when EF was used as query.

Conclusion

I have cloned, expressed, isolated and characterized a Bigelowiella protein with all characteristics of an α-actinin. Structural and functional annotation by Superfamily (Wilson et al., 2009) identified a N-terminal calponin-homology domain, a C-terminal EF-hand domain and a single spectrin repeat between the terminals. Direct binding assays as well as transmission electron microscopy showed that the protein cross-links actin filaments into dense bundles. Although experiments to localize the actin-binding site were impossible due to the protein’s propensity to precipitate, it is very likely that binding occurs to the N-terminal and the calponin homology domain, as in other α-actinins. Since both the full-length protein and the two C-terminal constructs bound calcium, it is apparent that the calcium-affinity is due to one (or more) of the EF-hand motifs. Since only the first EF-hand motif has the proper residues required for coordination of the calcium ion, it is probable that this motif also is responsible for the calcium binding, similar to the calmodulin-like domain of human α-actinin1 (Drmota Prebil et al., 2016).

Although some residues occurred in the disallowed regions of the Ramachandran plots, the quality of the predicted structures were high and very similar to known α-actinin domain structures. Major differences in all domain models were located to loop regions.

Based on the experimental results as well as on the homology modelling, is it evident that this Bigelowiella protein is a genuine α-actinin.

It has been suggested that the rod domain in α-actinin may function as a platform coordinating interactions with structural and signalling proteins (Djinovic-Carugo et al., 2002). Since the rod domain in Bigelowiella α-actinin spans only one spectrin repeat, in contrast to four repeats in α-actinins from multicellular organisms, less surface area is available for such a function. Alpha-actinin has been implicated in several cellular functions, such as cytokinesis (Fujiwara, Porter & Pollard, 1978; Li et al., 2016), movement (Matsudaira, 1994; Meacci et al., 2016; Sobue & Kanda, 1989) and contraction (Gautel & Djinovic-Carugo, 2016; Maruyama & Ebashi, 1965). Whether Bigelowiella α-actinin has similar functions requires further studies.

Supplemental Information

Figure S1 Low speed actin co-sedimentation assay

Bigelowiella α-actinin-like protein and polymerised actin were incubated for 30 min separately or together at different concentrations. After 30 min at room temperature, the samples were centrifuged (13,000 rpm, 15 min) and supernatant (S) and pellet (P) were separated on 10% SDS-PAGE and visualized by Coomassie Blue staining. Lane 1: 5 µM actin; Lane 2: 30 µM α-actinin-like protein; lanes 3–11: 5 µM actin together with 30, 20, 15, 10, 5, 10, 5, 2.5 and 1.25 µM α-actinin-like protein; M: protein ladder.

Click here for additional data file.

Supplemental Information 1 Model of Bigelowiella alpha-actinin actin-binding domain

Click here for additional data file.

Supplemental Information 2 Model of Bigelowiella alpha-actinin calmodulin-like domain

Click here for additional data file.

Supplemental Information 3 Model of Bigelowiella rod domain

Click here for additional data file.

Supplemental Information 4 Model of Bigelowiella alpha-actinin calmodulin-like domain (short version)

Click here for additional data file.

Facilities were provided by Umeå Core Facility for Electron Microscopy UCEM, National Microscopy Infrastructure, NMI (VR-RFI 2016-00968) and the Protein Expertise Platform (PEP).

Additional Information and Declarations

Competing Interests

Author Contributions

Data Availability

The author declares there are no competing interests.

Lars Backman conceived and designed the experiments, performed the experiments, analyzed the data, contributed reagents/materials/analysis tools, wrote the paper, prepared figures and/or tables, reviewed drafts of the paper.

The following information was supplied regarding data availability:

The pdb models are provided as Supplemental Files.

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
