# Peer review of "Alpha-actinin of the chlorarchiniophyte Bigelowiella natans"

_PeerJ, doi:10.7717/peerj.4288_

## Round 0.1 · original submission · Major Revisions

The paper was carefully reviewed by two experts in the field, and I agree that overall the experiments are executed carefully and interpreted appropriately. However, I also agree that upon re-submission the number of times that each experiment was performed has to be stated (and of course N should not equal 1) to demonstrate that experiments have been repeated a sufficient number of times to be confident of the reproducibility of the data. Another important issue raised by one of the reviewers is the requirement to demonstrate dimerisation (as against general aggregation that could compromise the interpretation of the data in Figs 6 and 7). Both mentioned concerns are sufficiently important to justify my recommendation "major revision". In your resubmission, please address also the minor points raised by the reviewers.

Reviewer 1 ·

Basic reporting

In this manuscript, Backman follows up his previous work on microbial alpha-actinin. This paper analyses the first example of an alpha-actinin from a photosynthetic organism that I'm aware of. The aim of the paper is well stated and generally the manuscript is written in a clear and engaging manner. The manuscript is well structured and well presented.

In terms of reporting the current data, I have some points for the author.

1.1 It would be helpful to include a table of Pfam and Smart E-values for the domain assignments alongside the Superfamily data already given in the text. The spectrin triple helix in the rod domain is clearly very divergent, as the author indicates. It is not recognised by Pfam or Smart as far as I can tell, and this is a worthwhile point to bring out.

1.2 On the same issue, I think it would be appropriate to put in a sequence alignment of the triple helix in the rod domain with those of Entamoeba histolytica, and S.pombe alpha-actinin, and representative repeats of one (or a few) four-repeat actinins.

1.3 Am I right in thinking that this is the first alpha-actinin that has just one spectrin repeat to be experimentally characterized? If so, I think this should be brought out more clearly in the text. This is particularly important in terms of the mechanism of dimerisation. In other alpha-actinins, dimerisation occurs by interaction of R1 with R2 or R4 (depending on the number of repeats). This alpha-actinin presumably dimerises by homotypic interactions of a pair of R1-equivalents. This should be stated clearly, and used to qualify the abstract (line 29:"all common characteristics of a typical alpha-actinin).

1.4. It would be helpful to include a diagram to indicate where Bigelowiella sits in eukaryotic phylogeny, and how its structure relates to other alpha-actinins.

Experimental design

The experiments are designed generally very well and reported in a way that would enable reproduction by another worker.

2.1. It would be helpful to explain a little further how the assignments for the beginning and end of each domain construct were made. I assume these were essentially from the superfamily assignments? The construction of the rod domain is reasonable, but I wondered if the author attempted to make other constructs of the triple helix that would be soluble? It's generally the case that triple helical repeats can be prepared in soluble form providing the phasing of the repeat is correct and there is a minimal number of residues at the end of the construct. In that sense I am not surprised that the rod domain construct was insoluble.

Major point
2.2. It is characteristic of alpha-actinins that they dimerise. The author correctly states that a dimeric alpha-actinin should bundle actin filaments, and this is indeed observed. But it does not prove that this alpha-actinin is a dimer. In order to substantiate the author's contention that this alpha-actinin is indeed a dimer (e.g. line 26 - "probably forms anti-parallel dimers"), I think it is important that this should be directly investigated. The gel filtration data in figure 2 indicates that unaggregated alpha-actinin can be obtained from the column. In my view, the author should analyse this for the oligomeric state of the protein. There are numerous biophysical methods that could be used for this, or alternatively chemical cross-linking and gel electrophoresis. In this context, given that the alpha-actinin has a tendency to aggregate, the results shown in figure 6 and figure 7 for bundling actin could result from aggregation of the protein rather than specific dimerisation. The author is correct to point to the tendency of the full length alpha-actinin to aggregate in the conditions used for the bundling assays, therefore the dimerisation is not proven.

Validity of the findings

Subject to my concerns in point 2.2, the data seem entirely valid. The protein clearly has functional actin-binding and Ca2+-binidng domains, characteristic of alpha-actinins. The structural modelling of the domains seems good (and for my own interst, I obtained the same results using Phyre).

Additional comments

Overall, I think this is a generally very helpful study of a microbial alpha-actinin from a novel class of organism. The author is to be commended on the clarity of the paper and the direct and straightforward experimental design. However, I think the author should directly address my point 2.2 before this paper could be published.

Reviewer 2 ·

Basic reporting

With the exception of a few minor errors the article is well written and referenced and the data are appropriately presented.

Experimental design

The research question is well defined and the experimental design is appropriate and well documented.

Validity of the findings

The number of times that each experiment was performed needs to be stated in each figure legend. This is important so that the reader can easily assess the reproducibility of the data.

Statistical analysis has been performed where appropriate.

Additional comments

The manuscript represents a thorough analysis of the a-actinin like protein from Bigelowiella natans. Given the interesting phylogeny of this organism, this analysis has some significance in terms of understanding the evolution of the actinin family of actin cross linking proteins. Overall the experiments are executed carefully and interpreted appropriately.

I would recommend publication upon completion of the following minor revisions.

Line 76 typographical error: isolatiOn
Line124/125 reference needed for sentence ending …. as before.
Line 199/200/201 This sentence is repeated in Line 212/213 – it seems a bit redundant to describe this in both places.
Line 219 change reliable to reliably.
Line 333 change a-actinin to alpha-actinin
Legend for figure 2: Use rcf rather than rpm when describing centrifugation conditions
Each figure legend containing experimental data should state the number of times that the experiment was performed. For example: “Data shown is representative of 3 independent experiments.” This is important so that the reader can easily assess the reproducibility of the data.
Figure 6: It would be useful to present the quantification of the co-sedimentation data. For example, plotting the proportion of actin in the pellet against actinin concentration in a line graph.

---

## Round 0.2 · accepted · Accept

Thanks for having done a great job in revising the manuscript! As you can see from reviewer 2's comments, they feel that all major concerns raised by him have been addressed properly. As reviewer 1 was not able to re-review this time, I myself went carefully through your rebuttal letter and the revised version and feel that also all points raised by reviewer 1, especially the major point with respect to dimerization, were addressed with great care and in my opinion sufficiently.

Reviewer 2 ·

Basic reporting

OK

Experimental design

OK

Validity of the findings

OK

Additional comments

The issues raised in my review of the manuscript have been adequately addressed in the revised version.